# Kinetic Analysis for High-Temperature Coarsening of γ″ Phase in Ni-Based Superalloy GH4169

**DOI:** 10.3390/ma12132096

**Published:** 2019-06-28

**Authors:** Cheng Zhang, Liming Yu, Hui Wang

**Affiliations:** 1School of Materials and Engineering, North China Institute of Aerospace Engineering, Langfang 065000, China; 2School of Materials Science & Engineering, Tianjin University, Tianjin 300354, China; 3Science and Technology on Reactor Fuel and Materials Laboratory, Nuclear Power Institute of China, Chengdu, Sichuan 610041, China

**Keywords:** γ″ phase, GH4169, aging, coarsening kinetics, Ni-based superalloy

## Abstract

The growth of precipitates in Ni-based superalloy GH4169 is critical as it controls the mechanical properties and long-term stability of the alloy. In this paper, the coarsening behavior of the main strengthening phase γ″ in the temperature range from 800 °C to 900 °C is investigated. Two heat treatment steps, i.e., pre-precipitation of γ″ phase and coarsening of precipitates at high temperatures, were performed on the GH4169 alloy. It was found that there were three morphological forms of γ″ phase in chronological order: lip-shape, disc-shape, and irregular rectangle-shape with larger size. The coarsening kinetics of γ″ phase followed the Lifshitz-Slyozov-Wagner (LSW) time-law for diffusion-controlled growth, and the activation energies of γ″ phase before and after losing the coherent relationship with matrix were 261 kJ mol^−1^ and 271 kJ mol^−1^, respectively.

## 1. Introduction

GH4169 (US brand name Inconel 718) is a Ni-based superalloy that has been extensively used under high temperature environment such as turbine engine, cogeneration, and heat treatment equipments. Its excellent high-temperature performance, especially the mechanical properties, is significantly affected by various precipitates [1,2,3]. Previous studies have shown that there are three major precipitates in GH4169: equilibrium γ′ phase (Ni_3_(Al,Ti) composition and L1_2_ structure), equilibrium δ phase (Ni_3_Nb composition and D0a structure), and metastable γ″ phase (Ni_3_Nb composition and D0_22_ structure) [4]. Both the spherical-shaped γ′ and the disc-shaped γ″ contribute to the age-hardening process of GH4169. However, as the major strengthening phase, γ″ plays a much more important role than γ′ due to the coherent strain and disordering effect [5,6,7]. Meanwhile, the stability of γ″ phase is also of great importance to the stability of GH4169, especially at high temperature, because the unfavorable phase transformation from γ″ to δ can lead to sharp degradation of the strength and hardness [8,9,10,11,12,13]. 

In the past decades, γ″ phase has attracted much attention worldwide, and relevant studies involve its volume fraction, particle size, long-term stability, and coarsening rate in the Ni-based alloys. Han et al. [14] found that the volume fraction ratio between γ′ phase and γ″ phase ranged between 2.5 and 4. These two precipitates in Inconel 718 alloy followed the time-law prediction of the Lifshitz-Slyozov-Wagner (LSW) theory [15,16] when aging between 670 °C and 730 °C. Wang et al. [17] compared the coarsening process of γ″ phase with and without electric treatment at 750 °C and 800 °C. They discovered that electric treatment can decrease the growth activation energy of γ″ phase. Besides, Devaux et al. [18] calculated the growth activation energy of γ″ phase and the interfacial energy between γ″ phase and matrix through isothermal and isochronal aging in the temperature range from 650 °C to 750 °C. It is noted that the previous studies mainly focused on the behavior of γ″ phase below 800 °C, and seldom investigated the evolution behavior of γ″ phase at higher temperatures. However, the stabilization and coarsening of γ″ phase above 800 °C is critical to understanding the high temperature thermal properties of GH4169. 

In this work, we are primarily interested in (1) the coarsening behavior of γ″ phase between 800 °C and 900 °C across various aging times; (2) the growth activation energies of γ″ phase before and after losing the coherent relationship with the matrix; and (3) a better understanding of the precipitation behavior of δ phase. The third interest stems from the temperature overlap of δ phase precipitation with γ″ phase coarsening. At the same time, the metastable γ″ phase can transform to the stable δ phase during the high-temperature aging process, resulting in the degradation of the alloy properties.

## 2. Materials and Methods 

The Ni-based alloy GH4169 in this study was provided by Baosteel Special Materials Corporation (Shanghai, China), with the chemical composition given in Table 1. The samples were machined into test blocks with dimensions of 10 mm × 10 mm × 10 mm, and followed by a homogeneous solution treatment at 1050 °C for 1 h to remove most precipitates, as well as residual stress during the rolling process. The starting structure of samples only consists of equiaxed austenite grains and very small amounts of carbides.

A total of 18 samples were investigated after a two-step heat treatment. The first step was an aging treatment at 720 °C for 4 h with subsequent air cooling to pre-precipitate a significant fraction of γ″ phase. Afterward, the samples were aged at three selected temperatures (i.e., 800 °C, 850 °C, and 900 °C) for various times (i.e., 0.1 h, 1 h, 2 h, 5 h, 8 h, 11 h) and then cooled in air. Thus the morphological changes and the coarsening behavior of γ″ phase were investigated in detail.

The microstructure of the heat-treated alloys was characterized by optical microscopy (OM, BX41M, Olympus, Tokyo, Japan) and scanning electron microscopy (SEM, S-4800, Hitachi, Tokyo, Japan). All specimens were prepared with standard grinding and polishing procedures, and then etched chemically with a solution consisting of CuCl_2_ 5 g + HCl 100 mL + C_2_H_5_OH 100 mL at room temperature for 7–9 min. The content of δ phase and the size of γ″ phase were measured quantitatively from the OM or SEM images using the Image-Pro-Plus 6.0 analysis software. For transmission electronic microscopy (TEM, JEM-2100F, JEOL, Tokyo, Japan) observations, thin slices of 300 μm in thickness were first cut from the specimens, then mechanically polished to thin foils of 50 μm in thickness, and finally electrochemically thinned by a twin-jet polisher in a 5% perchloric acid + 95% alcohol solution. 

## 3. Results and Discussion

### 3.1. Precipitation of δ Phase

Figure 1 shows the OM images of the specimens aged at 800 °C, 850 °C, and 900 °C for various times. At 800 °C, the precipitation rate of δ phase is extremely slow and only a small amount of round particles precipitate at grain boundaries (Figure 1a,b). Due to the small volume fraction, it is not easy to notice microstructure changes during the aging treatment. In contrast, when aging at 850 °C, the grain and twin boundaries become blurred due to the precipitation of δ phase at these locations, as denoted by the arrows in Figure 1d. The most obvious changes occur at 900 °C. The arrows in Figure 1f show that long strips of δ phase grow throughout both the grains and the boundaries in parallel when the aging time reaches 11 h. 

The precipitation of δ phase during the aging process begins with the nucleation of small particles at grain or twin boundaries, as shown in Figure 2a. This is because the structural defects at grain and twin boundaries can help to minimize the non-chemical free energy requirements for the nucleation process. These small particles then grow into paralleled platelets and invade into the adjacent grains, as shown in Figure 2b. Previous studies have found that δ phase can also nucleate from the paralleled dislocations in matrix or the faults within γ″ phase, especially along the close packed (112)γ″ planes [19,20,21]. This intra-granular precipitation of δ phase usually requires a longer incubation time than the precipitation at grain or twin boundaries. Along with the fast coarsening, δ phase can absorb γ″ particles located on its growing path, as shown by the arrows in Figure 2c. Meanwhile, due to the rapid growth rate and heterogeneous diffusion, the boundaries of δ platelets are not smooth.

In order to further study the precipitation behavior of δ phase, we measured the area fractions at the above three selected temperatures. As shown in Figure 3, all the area fractions of δ phase increased with aging time. At the same time, when aging at 850 °C and 900 °C, changing process of the growth rate can be divided into two stages, which correspond to the grain/twin boundary precipitation and intra-granular precipitation respectively. A similar result has been reported by Beaubois et al. [19] in Inconel 718 alloy. It was found that, in the initial stage, the ratio of area fractions of δ phase in the materials with and without solution treatment was equal to the ratio of grain sizes of the materials; the first stage of gradual growth lasted for 3 h when aging at 850 °C and 900 °C. In this study, it is found that the amount of δ phase increases with the increase in temperature. In addition, the growth time of the initial stage extends to 5 h at 900 °C and 8 h at 850 °C. This is due to that the pre-precipitated γ″ phase occupies some nucleation sites and consumes lots of Nb atoms, thus shifting the initial nucleation and growth process of δ phase to a longer time. In the stage of intra-granular precipitation, Sundararaman et al. [21] showed that stacking faults which were considered as the nucleation sites of δ phase, usually appeared in γ″ precipitates with size larger than 100 nm. In this study, such faults are found when the size of the precipitated phase is more than 200 nm. Meanwhile, a large number of γ″ particles larger than 400 nm can still exist stably. Therefore, the intra-granular precipitation of δ phase requires a longer incubation time. It is worth noting that although it is difficult for the intra-granular δ phase to nucleate, it grows rapidly. After aging at 900 °C for 11 h, the area fraction of precipitation can reach to 12%, which is close to the result of Azadian et al. [22] in the ring-rolled alloy (approximately 10%), but significantly higher than the data of Beaubois et al. [19] in the solution-treated alloy (only 5%). This is because a large number of γ″ particles precipitate during the pre-aging process, and the dissolution of γ″ particles can promote the growth of δ phase, which has similar effects to the rolling process. The dissolved γ″ phase not only provides defects favorable for phase transition, but also facilitates the diffusion of Nb atoms.

The time lag between these two stages of precipitation can be used to control the position and the amount of δ phase. Since moderate fractions of δ phase at grain boundaries can improve the fatigue life time during solution treatment as well as provide desirable resistance to grain boundary creep fracture [23,24], reasonable use of this time lag can achieve better high temperature performance in GH4169.

### 3.2. Morphological Changes of γ″ Phase

TEM images of the samples after isothermal aging at 900 °C for various times were selected to study the morphological evolution of γ″ phase. Figure 4a shows a typical structure of γ″ phase aged at 900 °C for 2 h. The morphology presents a lip-shape consisting of two long lobes, and the existence of strain field around the lobes (as denoted by the arrows) indicates the coherent relationship between γ″ phase and the matrix. Figure 4b shows the corresponding selected area diffraction pattern, which confirms that these particles belong to [010] and [100] γ″ variations. When the size of the particles reaches to 200 nm, the morphology of γ" phase turns to be a disc-shape (as shown in Figure 4c). Two closely spaced particles may join together to form a "neck" or an L-shaped structure, as indicated by the arrows in Figure 4d. This “encounter” mechanism was first proposed by Davies et al. [25] in the Ni–Co–Al system. They reported that Ni_3_Al precipitates can coalesce in these two forms, which results in the broadening of precipitate size distribution. Sundararaman et al. [21] and Zhang et al. [26] only found the "neck" shaped particles in their research, and they proposed that the composite structure was directional. Moore et al. [27] supported this viewpoint, and pointed out that those particles were mainly connected by the long axis due to the variation of misfit strain. In our study, both connection forms are found, but it is difficult to clarify whether these composite structures are originated from physical associations or image overlaps. These complex morphologies would be further observed in subsequent coarsening studies, and the results confirm that the precipitated phase is more prone to long-axis-oriented connection.

When the aging time reaches to 8 h at 900 °C stacking faults within γ″ phase can be observed in the TEM images. Some γ″ particles are separated by these stacking faults. One side of these particles remains in its original shape while the other side grows abnormally. The abnormal growth can result in a disc-shaped morphology, as shown in Figure 5a, or a trapezoid one, as shown in Figure 5b. With further increase of the size, stacking faults occur multiple times within the same particle, and the morphology of γ″ phase becomes an irregular rectangle-shape as shown in Figure 5c. In addition, some stacking faults extend from the particles into the matrix, which can be the nucleation site of δ phase, as indicated by the arrows in Figure 5d.

### 3.3. Coarsening Kinetics of γ″ Precipitates

Figure 6 shows the SEM images of samples aged at various temperatures and for various times. Since γ′ particles have mostly dissolved in the high-temperature aging process, the coarsening behavior can be considered as the result of γ″ particles only. After aging at 850 °C for 11 h, as can be seen clearly from the arrows in Figure 6d, the composite structures of γ″ phase are mainly connected in the long axis direction whereas the L-shaped connection is rare due to its inability of stable existence and growth. In the sample aged at 900 °C for 11 h, the quantity of γ″ particles decreases owing to the Ostwald ripening process. In addition, the edges of precipitates become irregular due to the abnormal growth of γ″ particles, as shown in Figure 6f.

The basis of the "directional encounter" described above is the functional relationship between the aspect ratio *q* and the radius of the disc-shaped particle *L*. Overall, the aspect ratio decreases with the coarsening process of γ″ phase, but the slope change is not continuous. In this regard, Devaux et al. [18] summarized his own research and previous researches, and pointed out that γ″ phase lost the coherent relationship with the matrix when the size of the precipitated phase reached to 35–50 nm in the aging process between 650 °C and 750 °C. This conclusion was confirmed by Zhang’s study [26], but the size change point was determined as 150 nm in the aging process between 750 °C and 850 °C. It is noticed that few data points can be found in previous studies; meanwhile, the conclusions of different researchers are likely to be affected by different alloy compositions and aging temperatures. Therefore, we statistically calculated the shape change of a large number of γ″ precipitates at the three selected temperatures, and provided the acquired data points and fitting results as Figure 7 shows.

Due to the "encounter" behavior and the abnormal growth of γ″ particles, the data points appeared to be scattered. Recently, in the study of Alloy 625, Moore et al. [27] attributed this data scattering to the gradual continuous transformation of particle morphology and fitted the data points with a hyperbolic curve. However, this fitting method cannot explain the morphology difference of the precipitated phase during the isothermal processes at different temperatures. In view of this, this study adopted a piecewise linear fitting method to determine the turning points of morphological changes. Since the aging time at 800 °C is too short to fit the turning point, only a decreasing trend of the aspect value *q* can be observed in Figure 7a. Figure 7b,c shows that the piecewise linear fitting curves intersect at the particle sizes of 136 nm and 176 nm respectively. That is to say, as the temperature increases, the coherent relationship can be maintained to larger particle size which is probably because higher aging temperatures can release more stress. At the same time, it is found that the coherence and non-coherence transition is prone to occur when the *q* value is about 0.3.

The LSW theory of volume diffusion-controlled growth is often used to describe the coarsening process. The kinetic equation for a spherical precipitated particle can be expressed as follows [15,16,28]:(1)$$r^{3}-r_{0}{}^{3}=\frac{8}{9}\frac{\Gamma C_{e}V_{m}{}^{2}D}{RT}t=Kt$$
where *r* is the average radius of the growing particles at time *t*, *r*_0_ is the radius at the onset of the coarsening process, Γ is the interfacial free energy at the particle and matrix interface, *D* is the diffusion coefficient of solute atoms in the matrix, *C_e_* is the concentration of solute atoms in equilibrium with a particle of infinite radius, *R* is the Boltzmann constant, *T* is the temperature of aging heat treatment, and *V_m_* is the molar volume of precipitates. 

This equation has been modified by Boyd et al. [29] for disc-shaped particles in Al-Cu alloys:(2)$$L^{3}-L_{0}{}^{3}=\frac{128}{9q\pi }\frac{\Gamma C_{e}V_{m}{}^{2}D}{RT}t=K^{\prime \prime }t$$

According to the LSW theory, if the coarsening process is controlled by volume diffusion, the corresponding relationship plot of *L*^3^–*L*_0_^3^ versus *t* should be a straight line. Figure 8 shows the evolution of the diameter of γ″ precipitates as a function of aging time at 800 °C, 850 °C, and 900 °C. As γ″ phase is very fine after pre-aging at 720 °C for 4 h, it is considered that the coarsening process starts from the subsequent aging process at higher temperature. Thus, the value of *L*_0_ is assumed to be zero. As can be seen from Figure 8, there is a linear relationship between *L*^3^–*L*_0_^3^ and *t* at 800 °C, but the linear relationship is segmented when aging at 850 °C and 900 °C. Other researchers also found this segmentation, but they attributed it to the experimental data scattering or the faulty assumption [18,25,30,31]. Table 2 lists the slope of each straight line. It can be seen that the coarsening rate *K*″ obviously increases when the particle size increases up to 122 nm at 850 °C and 181 nm at 900 °C. It is worth noting that the transition points of *K*″ are very close to the coherent transition sizes, which indicates that the coherence and non-coherence transition leads to the transformation of coarsening kinetics. In general, the coarsening process of γ″ phase both before and after such transition follows the LSW theory.

The diffusion coefficient *D* in Equation (2) is defined as:(3)$$D=D_{0}\exp(\frac{-Q_{\gamma \prime \prime }}{RT})$$

*K*″ can thus be deduced from Equation (3) as follows: (4)$$K^{\prime \prime }=\frac{128}{9q\pi }\frac{\Gamma C_{e}V_{m}{}^{2}D_{0}}{RT}\exp(\frac{-Q_{\gamma "}}{RT})$$

Assuming that the concentration *C_e_* is a constant over the temperature range, the aspect ratio *q* takes the average value of all experimental data and can also be viewed as a constant. Consequently, the activation energy *Q_γ″_* for the coarsening of γ″ precipitates can be easily determined from the slope of ln(*K″T*) versus 1/*T* plot. Figure 9 illustrates the activation energies of γ″ phase before and after losing the coherent relationship with the matrix. By linear fitting, the activation energies can be determined as 261 kJ mol^−1^ and 271 kJ mol^−1^, respectively. This slight increase of activation energy may be attributed to the movement difficulty of vacancies after losing the coherent relationship between the γ″ phase and the matrix. Compared with 285.6 kJ mol^−1^ reported by Wang et al. [17] and 292 kJmol^−1^ reported by Zhang et al. [26], the values in this study are much more close to the activation energy of Nb diffusion in nickel-based alloys (257 kJ mol^−1^) [18]. On the one hand, the pre-precipitated γ″ phase can suppress the precipitation of δ phase and Nb atoms in the matrix and are mainly used for the growth of γ″ phase in the initial stage of the coarsening process, so the measured diffusion activation energy is lower than that of the simultaneous precipitation of both γ″ phase and δ phase. On the other hand, the differences in alloy composition and the selected heat treatment time nodes can also result in some changes in the diffusion activation energy. To sum up, our study demonstrates that the coarsening process of γ″ phase is controlled by the volume diffusion of Nb atoms, and follows the LSW theory before and after the coherence and non-coherence transition.

## 4. Conclusions

The following conclusions regarding the coarsening kinetics of γ″ phase in Ni-based superalloy GH4169 can be drawn on the basis of this study:(1)The grain/twin boundary precipitation and intra-granular precipitation of δ phase both occur during the aging processes at 850 °C and 900 °C. There is a time lag between the grain/twin boundary precipitation and the intra-granular precipitation of δ phase, and the pre-precipitating γ″ phase can suppress the intra-granular precipitation of δ phase.(2)The morphology of γ″ phase changes from a lip-shape to a disc-shape and finally becomes an irregular rectangle-shape. (3)The coarsening process of γ″ phase in GH4169 is controlled by the volume diffusion of Nb atoms in the matrix and follows the LSW theory. The activation energies before and after γ″ phase losing the coherent relationship with matrix are determined as 261 kJ mol^−1^ and 271 kJ mol^−1^, respectively. 

## Figures and Tables

**Figure 1 materials-12-02096-f001:**
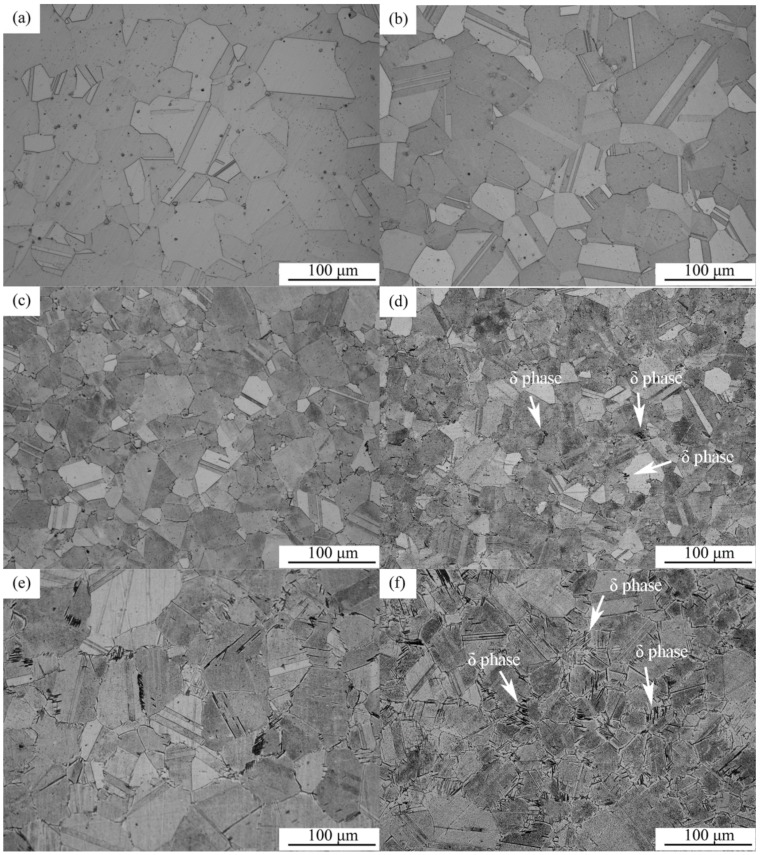
Optical microscopy (OM) images of GH4169 alloy after isothermal aging at (**a**) 800 °C for 5 h; (**b**) 800 °C for 11 h; (**c**) 850 °C for 5 h; (**d**) 850 °C for 11 h; (**e**) 900 °C for 5 h; and (**f**) 900 °C for 11 h.

**Figure 2 materials-12-02096-f002:**
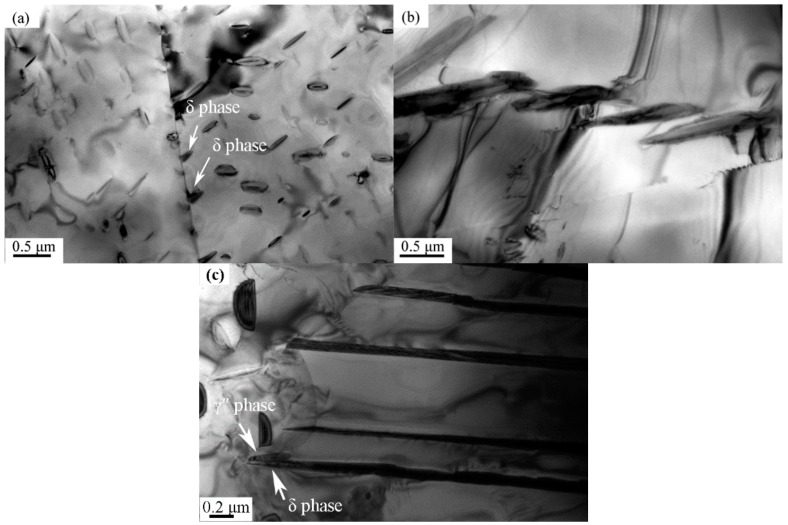
Transmission electronic microscopy (TEM) images of (**a**) δ phase precipitated at grain boundaries; (**b**) paralleled δ phase close to the grain boundaries; (**c**) the encounter between δ phase and γ″ phase in GH4169 alloy.

**Figure 3 materials-12-02096-f003:**
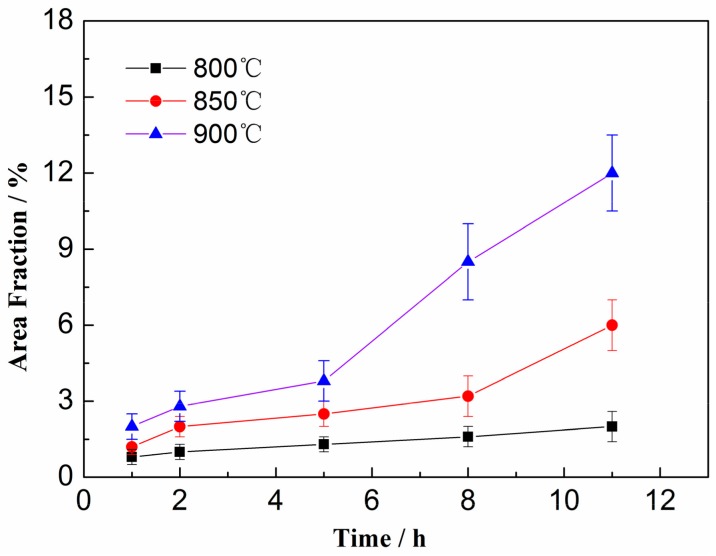
Area fractions of δ phase versus aging time at different selected temperatures.

**Figure 4 materials-12-02096-f004:**
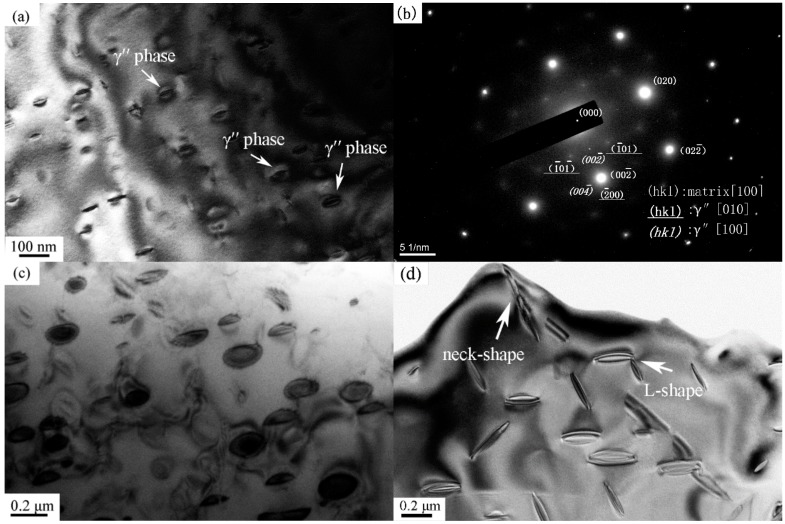
TEM images of (**a**) lip-shaped γ″ phase after isothermal aging at 900 °C for 2 h; (**b**) selected area diffraction pattern of γ″ phase; (**c**) disc-shaped γ″ phase after isothermal aging at 900 °C for 5 h; (**d**) "neck" and L-shaped structures of γ″ phase after isothermal aging at 900 °C for 5 h.

**Figure 5 materials-12-02096-f005:**
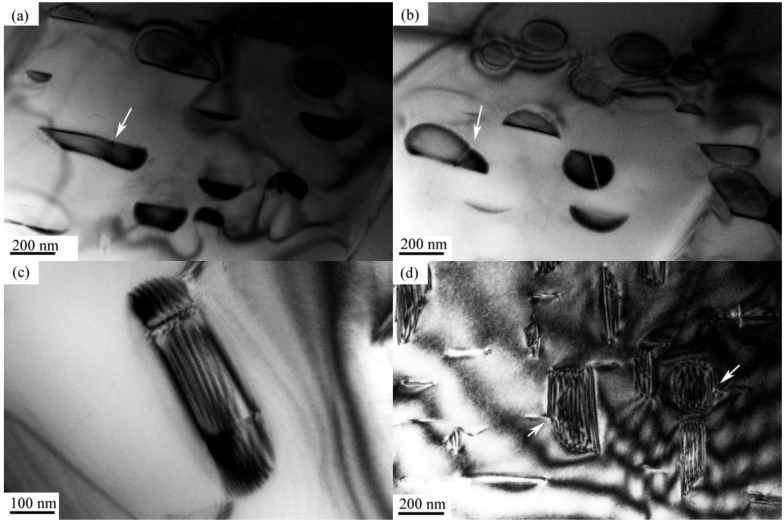
TEM images of (**a**) trapezoidal abnormal growth of γ″ phase after isothermal aging at 900 °C for 8 h; (**b**) disc-shaped abnormal growth of γ″ phase after isothermal aging at 900 °C for 8 h; (**c**) planar faults within irregular rectangle-shaped γ″ phase after isothermal aging at 900 °C for 11 h; (**d**) planar faults extending from γ″ phase to matrix after isothermal aging at 900 °C for 11 h.

**Figure 6 materials-12-02096-f006:**
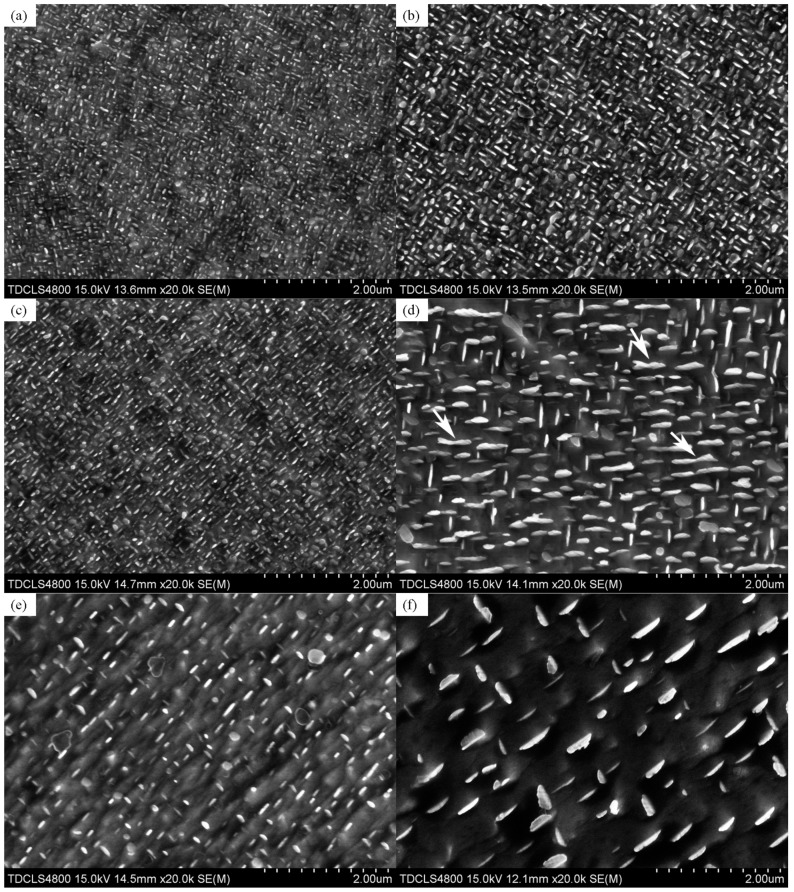
SEM images of the explored GH4169 alloy after isothermal aging at (**a**) 800 °C for 2 h; (**b**) 800 °C for 11 h; (**c**) 850 °C for 2 h; (**d**) 850 °C for 11 h; (**e**) 900 °C for 2 h; and (**f**) 900 °C for 11 h.

**Figure 7 materials-12-02096-f007:**
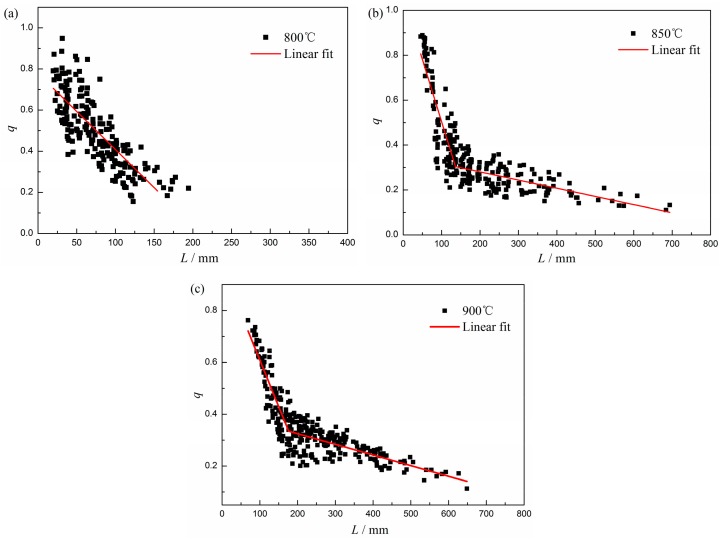
Variation in the aspect ratio *q* (*q* = *e/L*, where *e* is the thickness of the disc-shaped particle) versus the radius of disc-shaped particle *L* in GH4169 after isothermal aging at (**a**) 800 °C; (**b**) 850 °C; and (**c**) 900 °C.

**Figure 8 materials-12-02096-f008:**
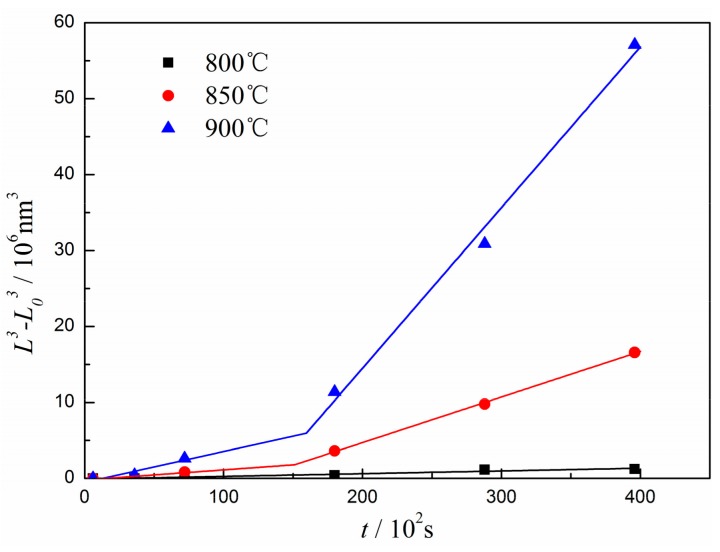
Evolution of the diameter of γ″ precipitates as a function of aging time *t*.

**Figure 9 materials-12-02096-f009:**
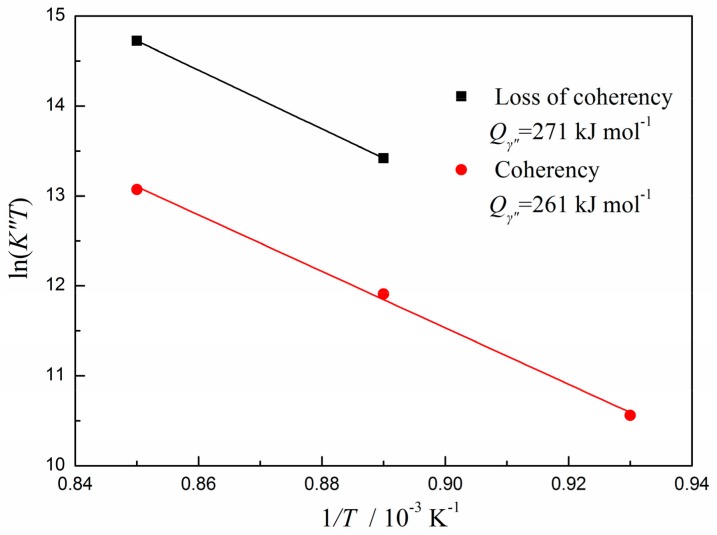
Determination of the activation energies of γ″ phase before and after the coherence and non-coherence transition in GH4169.

**Table 1 materials-12-02096-t001:** Chemical composition of the investigated GH4169 alloy (wt%).

**Element**	Ni	Cr	Nb	C	Mn	Si	Mo	Cu
**Content**	52.70	19.63	5.26	<0.08	<0.35	0.13	3.17	<0.30
**Element**	Co	Al	Ti	B	S	P	Fe	
**Content**	<1.00	0.63	1.01	<0.01	<0.01	<0.01	15.79.	

**Table 2 materials-12-02096-t002:** The determined *K*″ values in Equation (2).

Temperature (°C)	800	850	900
***K*″/(10 nm^3^ s^−1^)**	*K*″ = 3.60	*K*″ = 13.26 for *L* < 122 *K*″ = 60.00 for *L* > 122	*K*″ = 40.52 for *L* < 181 *K*″ = 211.46 for *L* > 181

Note: *K*″ is established on *L* (nm) range.

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
