# Peer review of "Kinetic Analysis for High-Temperature Coarsening of γ″ Phase in Ni-Based Superalloy GH4169"

_materials, 2019, doi:10.3390/ma12132096_

Round 1
Reviewer 1 Report
This manuscript is generally well-written and discussed. Authors have provided enough background in the introduction and the quality of SEM and TEM imaging is excellent. I recommend this paper for publication after addressing the following comments:
1- How consumption of Nb atoms increases the incubation time? Does Nb diffuse from gamma phase to delta phase? If this is the case, how Mo affects the precipitation behaviour of delta phase?
2- There are a few grammatical mistakes that need attention.
3- A very important missing element in figure 7 is time. Why time has not been taken into account and not reported in this figure.
Author Response
Dear Editor/Reviewer:
Thank you for your comments and suggestions concerning our manuscript. Those comments are all valuable and very helpful for revising and improving our paper. We have studied the comments carefully and have made some revisions according to the reviewer’s comments. Revised portion is marked in red in the revised manuscript. The following are our point-by-point responses to each of the comments. We hope that these revisions and our accompanying responses will be sufficient to make our manuscript suitable for publication.
Kind regards.
Dr. Liming Yu
Responds to the reviewer 1:
1. How consumption of Nb atoms increases the incubation time? Does Nb diffuse from gamma phase to delta phase? If this is the case, how Mo affects the precipitation behaviour of delta phase?
Response: Thank you very much for your comment.
(1) Both the γ″ phase and the δ phase are composed of Ni3Nb, but γ″ phase has a D022 crystal structure while δ phase has a D0a crystal structure. The intra-granular precipitation of δ phase usually nucleates from the paralleled dislocations in the matrix or the faults within γ″ phase. The pre-precipitated γ″ phase consumes lots of Nb atoms, which reduces the Nb content of the matrix and results in difficulty in δ phase precipitation. At the same time, the γ″ phase found in this study is more stable than previous studies, which means the nucleation of δ phase from the faults within γ″ phase also becomes difficult. In summary, we believe that the intra-granular precipitation of δ phase requires a longer incubation period.
(2) Since γ″ is a metastable phase and δ is a stable phase, Nb atoms diffuse from γ″ phase to δ phase during high temperature aging.
(3) In the GH4169 alloy, the addition of Mo can improve the properties of the material, but it has no effect on the precipitation of δ phase.
2. There are a few grammatical mistakes that need attention.
Response: We are sorry for the grammatical mistakes, and we have carefully revised the whole manuscript and corrected the grammar and spelling errors. All the changes have been marked in red in the revised manuscript. We believe that the language now is acceptable.
3. A very important missing element in figure 7 is time. Why time has not been taken into account and not reported in this figure.
Response: Thank you very much for your comment. In each of the graphs in Figure 7, we selected 250 data points (i.e., 50 data points each at 1 h, 2 h, 5 h, 8 h, 11 h), but the data points represented by these times are not distinguished in the figures. This is because, as a whole, the trend of q with L is the same with the trend of q with t, and we believe that the trend of q with L can better reflect the morphology changes. In addition, there are too many data points and many overlap each other. If all the data points are distinguished according to time, it will affect the visual experience of the figures.

Reviewer 2 Report
This paper deals with the structural characterization of a Ni-based superalloy. The morphology and growth features of the precipitates in the alloy upon aging treatment have been studied. In my opinion, the paper can be published after certain revision has been implemented.
I think that the chemical nature of the alloy should be mentioned in the title of the paper and in the Abstract to attract a broader audience of readers.
In the Abstract, please revise this phrase: "the activation energy of γ″ phase could be improved". How can the activation energy be "improved"?
Please describe the etching technique in more detail. What was the etching time?
Please provide XRD patterns of the alloy samples.
What is the phase composition of the initial and aged alloy samples and what is the chemical composition of the precipitates? This information will be useful for readers to better understand the chemistry of this alloy.
Author Response
Dear Editor/Reviewer:
Thank you for your comments and suggestions concerning our manuscript. Those comments are all valuable and very helpful for revising and improving our paper. We have studied the comments carefully and have made some revisions according to the reviewer’s comments. Revised portion is marked in red in the revised manuscript. The following are our point-by-point responses to each of the comments. We hope that these revisions and our accompanying responses will be sufficient to make our manuscript suitable for publication.
Kind regards.
Dr. Liming Yu
Responds to the reviewer 2:
1. The chemical nature of the alloy should be mentioned in the title of the paper and in the Abstract to attract a broader audience of readers.
Response: Thank you for the suggestion. We have added the phrase "Ni-based superalloy" in the title of the paper and in the Abstract to attract a broader audience of readers.
2. In the Abstract, please revise this phrase: "the activation energy of γ″ phase could be improved". How can the activation energy be "improved"?
Response: We are sorry to have used inappropriate phrase in the Abstract, and we have revised the phrase "the activation energy of γ″ phase could be improved" into "the activation energies of γ″ phase before and after losing the coherent relationship with matrix were 261 kJ mol−1 and 271 kJ mol−1, respectively".
3. Please describe the etching technique in more detail. What was the etching time?
Response: According to your comment, we have modified the description of etching technique and added the corrosion time. We change the sentence "All specimens were prepared ...C2H5OH 100 mL.'' into "All specimens were prepared with standard grinding and polishing procedures, and then etched chemically with a solution consisting of CuCl2 5 g + HCl 100 mL+ C2H5OH 100 mL at room temperature for 7-9 min."
4. Please provide XRD patterns of the alloy samples.
Response: Thank you for the suggestion. However, we believe that different phases can be clearly distinguished from SEM, TEM and OM images, and these observation methods are more intuitive to illustrate the morphology and quantity changes of precipitates in GH4169. In order to avoid repeated expression, the XRD patterns are not included in the manuscript.
5. What is the phase composition of the initial and aged alloy samples and what is the chemical composition of the precipitates? This information will be useful for readers to better understand the chemistry of this alloy.
Response: According to your comment, we have added the chemical composition of the precipitates in the manuscript. We change the sentence "Previous studies have shown that ... metastable γ″ phase (D022 structure) [4]." into "Previous studies have shown that there are three major precipitates in GH4169: equilibrium γ′ phase (Ni3(Al,Ti) composition and L12 structure), equilibrium δ phase (Ni3Nb composition and D0a structure) and metastable γ″ phase (Ni3Nb composition and D022 structure) [4]." In addition, we have also added the sentence "The starting structure of samples only consists of equiaxed austenite grains and very small amounts of carbides." to illustrate the phase composition of initial samples.

Reviewer 3 Report
The article reports about high temperature structural-behavior of a nickel-based superalloy. The description into evolution of the structure due to time-temperature seems interesting and quite well presented. Also 18 samples were investigated – which, from my point of view is beneficial for this research. In my opinion it is well-written paper. I can address a few comment to the authors:
General remark:
Providing the grade of the alloy “GH4169” is not clear for a reader. I think you should better characterize that alloy. Simply, please add in the end of the title, in abstract, keywords and conclusion sections the phrase “nickel based superalloy”.
Specific remarks:
Table 1. – I am worried about the fact that “Fe” is balanced while article describes “Ni” based alloy. Please consider my remark.
The rolling direction usually influences on samples structure (texture)? Please tell me if samples were machined from a plate (or rod?), what was the thickness of the plate? Was the texture observed in samples LOM cross-section photos?
The description of the quantity of the tested samples is not clear. I think that you should emphasize it in the text. Please add in the “materials and methods” section information about total count of the investigated samples.
Please comment in the article if any structural changes were observed for a short time of aging (for time: 0.1h)?
L96: Please improve the lower case of γ″, in the phrase: “packed (112)γ″ planes [19-21]”
In line 110 authors referee to Inconel 718 alloy – please comment on the chemical composition differences between GH4169 and 718 alloys.
All the figures with plots: I think that X and Y axis should be uniformly presented/described in the whole article – now you have got different styles in fig. 3, 7, 8 and 9 (once unit is in bracket once after slash). Please improve it according to Materials requirements.
In fig. 7 – I suggest to add in the caption information that ‘e – means thickness of disc shaped particle”. It improves the readability of the plots.
Author Response
Dear Editor/Reviewer:
Thank you for your comments and suggestions concerning our manuscript. Those comments are all valuable and very helpful for revising and improving our paper. We have studied the comments carefully and have made some revisions according to the reviewer’s comments. Revised portion is marked in red in the revised manuscript. The following are our point-by-point responses to each of the comments. We hope that these revisions and our accompanying responses will be sufficient to make our manuscript suitable for publication.
Kind regards.
Dr. Liming Yu
Responds to the reviewer 3:
1. Providing the grade of the alloy “GH4169” is not clear for a reader. I think you should better characterize that alloy. Simply, please add in the end of the title, in abstract, keywords and conclusion sections the phrase “nickel based superalloy”.
Response: According to your suggestion, we have added the phrase "Ni-based superalloy" in the title, abstract, keywords and conclusion sections. We hope that this will help readers understand this alloy better.
2. Table 1. – I am worried about the fact that “Fe” is balanced while article describes “Ni” based alloy. Please consider my remark.
Response: According to your comment, we have modified Table 1. to give the specific content of "Fe". Since the content of Ni in the alloy is the most, it is suitable that the alloy is expressed as a Ni-based superalloy.
3. The rolling direction usually influences on samples structure (texture)? Please tell me if samples were machined from a plate (or rod?), what was the thickness of the plate? Was the texture observed in samples LOM cross-section photos?
Response: Thank you for your comment. The as-received material was a hot-rolled bar with a diameter of 10 mm, and the rolling direction did influence on samples original structure. During the experiment, in order to avoid the influence of rolling texture on the results, the observation surfaces we selected were both perpendicular to the rolling direction. In addition, we did a homogeneous solution treatment at 1050℃ for 1 h to remove most precipitates, as well as residual stress during the rolling process.
4. The description of the quantity of the tested samples is not clear. I think that you should emphasize it in the text. Please add in the “materials and methods” section information about total count of the investigated samples.
Response: According to your comment, we have added the description of the quantity of samples in the "Materials and Methods" section, and the total number of samples in the experiment was 18.
5. Please comment in the article if any structural changes were observed for a short time of aging (for time: 0.1h)?
Response: Thank you for your comment. The 0.1 h aging time is short and the structure changes are not obvious, so we did not list the observations of OM, SEM and TEM images in the manuscript. However, in the subsequent analysis process, especially in Figures 8 and 9, the 0.1 h aging results have an important effect on the analysis. This is why the result of the 0.1h aging process is rarely mentioned in the manuscript.
6. L96: Please improve the lower case of γ″, in the phrase: “packed (112)γ″ planes [19-21]”
Response: We are sorry for the negligence of the format and have been modified it in the manuscript.
7. In line 110 authors referee to Inconel 718 alloy – please comment on the chemical composition differences between GH4169 and 718 alloys.
Response: Thank you for the suggestion. In fact, Inconel 718 and GH4169 are the same material, except that Inconel 718 is the US grade and GH4169 is the Chinese grade. We have added the phrase "(US brand name Inconel 718) "in the "Introduction" section of the manuscript for readers to understand better.
8. All the figures with plots: I think that X and Y axis should be uniformly presented/described in the whole article – now you have got different styles in fig. 3, 7, 8 and 9 (once unit is in bracket once after slash). Please improve it according to Materials requirements.
Response: We are sorry that we ignored the consistency of the text format in the images, and we have modified Figure 3, 7, 8, and 9 in the manuscript.
9. In fig. 7 – I suggest to add in the caption information that ‘e – means thickness of disc shaped particle”. It improves the readability of the plots.
Response: According to the comment, we have added the phrase "(q=e/L, where e means the thickness of the disc-shaped particle)" in the caption of figure 7.
